# Ceramic nanowelding

Liqiang Zhang[1], Yushu Tang[1], Qiuming Peng[2], Tingting Yang[2], Qiunan Liu[2], Yuecun Wang[3], Yongfeng Li[1], Congcong Du[2], Yong Sun[2], Lishan Cui[1], Fan Yang[1], Tongde Shen [2], Zhiwei Shan [3] & Jianyu Huang[2]

Ceramics possess high temperature resistance, extreme hardness, high chemical inertness and a lower density compared to metals, but there is currently no technology that can produce satisfactory joints in ceramic parts and preserve the excellent properties of the material. The lack of suitable joining techniques for ceramics is thus a major road block for their wider applications. Herein we report a technology to weld ceramic nanowires, with the mechanical strength of the weld stronger than that of the pristine nanowires. Using an advanced aberration-corrected environmental transmission electron microscope (ETEM) under a $CO_2$ environment, we achieved ceramic nanowelding through the chemical reaction $MgO + CO_2 \rightarrow MgCO_3$ by using porous MgO as the solder. We conducted not only nanowelding on MgO, CuO, and $V_2O_5$ nanowires and successfully tested them in tension, but also macroscopic welding on a ceramic material such as $SiO_2$, indicating the application potential of this technology in bottom-up ceramic tools and devices.

[1] State Key Laboratory of Heavy Oil Processing, and Department of Materials Science and Engineering, China University of Petroleum, Beijing 102249, China. [2] Nano Energy Center, State Key Lab Metastable Materials Science and Technology, Yanshan University, Qinhuangdao 066004, China. [3] Center for Advancing Materials Performance from the Nanoscale (CAMP-Nano) & Hysitron Applied Research Center in China (HARCC), State Key Laboratory for Mechanical Behavior of Materials, Xi'an Jiaotong University, Xi'an 710049, China. Correspondence and requests for materials should be addressed to Y.L. (email: yfli@cup.edu.cn) or to Z.S. (email: zwshan@mail.xjtu.edu.cn) or to J.H. (email: jhuang@ysu.edu.cn)

Although many nanomaterials have been fabricated[1–3], how to join them together to produce more complicated nanodevices[4, 5] by using nanowelding is still challenging[6, 7]. Welding at the nanoscale is crucial to build nanodevices via the bottom–up approach[8, 9]. In recent years, scientists have successfully realized the joining of individual low-dimensional nanostructure materials including carbon nanotubes (CNTs)/metal[10], CNTs/CNTs[11], and metal/metal[12, 13], by either cold welding[14], Joule heating[15], or applying voltage/current[16]. Each welding technique has its special advantages, such as high speed, low cost, no contamination, no damage or an excellent weld junction. However, the reported techniques are mainly restricted to the welding of either CNTs or metals, and the welding of ceramic nanomaterials is seldom reported. Till now, ceramic nanomaterials can only be welded by depositing metal Pt, Au, Sn on the junction through heating or focused ion beam (FIB)[17], which is not only expensive and complex, but also contaminates and even damages the sample during the welding process[18]. Additionally, the poor infiltration between ceramic and metal makes it difficult to form a good joint. Furthermore, joining ceramic nanomaterials by using metal causes serious internal stress due to large thermal expansion coefficients mismatch. As we know, an ideal welding technique usually selects the same type of material as the solder, which can preserve its original properties and morphology without causing compatibility problems. This goal is hard to achieve for ceramic welding given the high melting point and good insulation of ceramic materials. Additionally, ceramic nanomaterials usually possess high hardness, no deformability and low diffusivity, which all lead to the extreme difficulty of ceramic nanowelding[19]. Developing a technique by using a ceramic as the solder for welding ceramics is therefore the ultimate goal for the nanowelding of ceramics. Ceramic welding is important for making more complex structures that are impractical or impossible to make using one-step processing.

MgO is widely used as an adsorbent for the capture of greenhouse gas ($CO_2$ gas)[20]. It was recently found that electron or plasma irradiation can promote a chemical conversion of $CO_2$[21, 22], which inspired us to explore the possibility of using the electron beam (e-beam) to stimulate the MgO and $CO_2$ reaction for ceramic nanowelding applications.

We conducted experiments in an advanced Cs image corrected environmental transmission electron microscope (ETEM) under the flow of $CO_2$ gas. Under e-beam irradiation in the $CO_2$ environment, nano-MgO reacts with $CO_2$ quickly without any external heating source or current. The irradiated area becomes fluidic with gas bubbles erupting violently and continuously, mimicking the boiling of viscous gels. By using this technique, we welded MgO, CuO, and $V_2O_5$ nanowires in the ETEM and carried out successful in situ tensile tests on these nanowires. Compared with traditional welding technologies, this technology achieves a full ceramic nanoweld using a ceramic solder, which is simple, low cost, high speed and without contamination. Most importantly, the welding spots exhibit a remarkable tensile strength that is even greater than that of the pristine nanowires.

## Results

**In situ nanowelding of ceramic nanowires in an ETEM.** The nanowelding process is shown schematically in Fig. 1a. We conducted nanowelding of MgO nanowires by using a Pico-Femto transmission electron microscope and a scanning tunneling microscopy (TEM-STM) holder (Fig. 1b) in a Cs-corrected FEI Titan ETEM. MgO nanowires were first glued on two tungsten (W) tips by using silver epoxy (Fig. 1a). Subsequently two individual MgO nanowires were manipulated toward each other by the movable piezoelectrics-controlled STM probe to carry out the welding experiments (Fig. 1a). To test the feasibility of the ceramic nanowelding, we first attempted to weld one MgO nanowire onto the W tip (the 1st welding). The connection between the W tip and the MgO nanowire was made in a head-to-head geometry (Fig. 2a). When the two nanowires come into intimate contact, pure $CO_2$ gas was released into the ETEM chamber. As the pressure of the $CO_2$ in the chamber reached 200 Pa, we focused the e-beam onto the nanowire junction with a medium dose rate (100 e nm$^{-2}$ s$^{-1}$). The e-beam dose rate plays a significant role in the speed of the welding (Supplementary Fig. 1). It was observed that a large amount of highly mobile bubbles emerged in the interior of the irradiated MgO nanowire, and the number and size of the bubbles grew with time (Supplementary Fig. 2 and Supplementary Movie 1). With increasing bubble number and size, the nanowire became fluidic and flowed like a porous viscous liquid. Upon reducing the e-beam intensity, the W tip and the MgO nanowire were welded together instantly (Fig. 2b, j). Besides the head-to-head welding mode, the welding could be performed in a side-by-side mode as well (Supplementary Fig. 3). Once the welding process completed, the $CO_2$ was removed from the ETEM chamber. After that, we found that the MgO nanowire was firmly welded onto the W tip, and the as-welded MgO nanowire retained its original morphology (Fig. 2b). To test the quality of the welding, we pulled the nanowire backward and found that the nanowire broke near the right contact rather than from the welding joint (Fig. 2c), proving that the tensile strength of the weld is greater than that of the pristine MgO nanowire.

After the first welding (MgO welding onto the W tip), we welded a second MgO nanowire (2nd welding) to the first nanowire residue attached to the W tip. Similar joining behavior to the first welding was also obtained (Fig. 2d–f, k). We then pulled the two welded nanowires backward, and the nanowire broke near the right contact, proving again that the tensile strength of the welding joints 1 and 2 is greater than that of the pristine MgO nanowire. A third nanowire was welded to the two nanowires attached to the W tip (Fig. 2g–i, l). As we pulled this

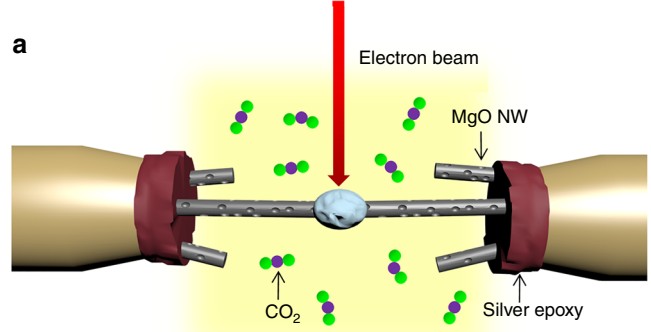

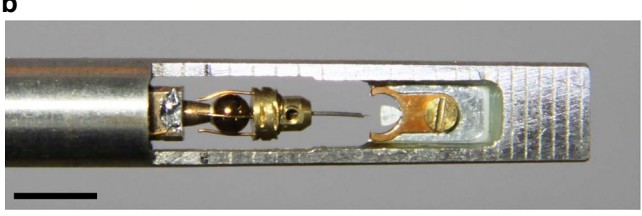

**Fig. 1** Schematics of the ceramic nanowelding setup. **a** The experiment was carried out in an ETEM with an atmosphere of $CO_2$. Prior to the experiment, several MgO nanowires were first glued to two Al probes using silver epoxy, and then were manipulated to approach each other using a STM holder. **b** The real image of the STM holder utilized in this study. Scare bar = 0.5 cm

nanowire backwards, it broke near the right contact of the third nanowire, not from the welding joints, proving again that the tensile strength of the welds is greater than that of the pristine MgO nanowire. Additionally, we did not see changes or slippage of the contact during all the tensile testing experiments, indicating the mechanical robustness of the contact.

Under the flow of $CO_2$ gas into the ETEM chamber and by focusing the e-beam to the nanowire junctions, nanowelding of ceramic nanowires was successful (Fig. 2, Supplementary Fig. 4 and Supplementary Movie 2). No extra heating or application of voltage/current is required, and the entire nanowelding process can be completed with both $CO_2$ and e-beam (Supplementary Fig. 5). All of the three weld junctions exhibit smooth surfaces without voids (Fig. 2j–l), representing the typical characteristics for a good weld. We have done identical welding tests for MgO nanowires with different diameters. There appears to be no limits in terms of the diameter for the welding of nanowires (Supplementary Fig. 6). Beside MgO nanowires, other types of ceramic nanowires with different diameters could also be welded together using this technique (Supplementary Fig. 7).

**Structure evolution of the nanowelds.** The welding process was monitored by TEM bright-field imaging, electron diffraction pattern (EDP) and electron energy loss spectroscopy (EELS) (Fig. 3). The pristine MgO nanowires fabricated through a hydrothermal method are porous and polycrystalline (Fig. 3a, e, i and Supplementary Fig. 8). Upon e-beam irradiation for several minutes (Fig. 3b), the original MgO gradually transformed from polycrystalline (Fig. 3a, e, i) into an amorphous structure (Fig. 3b, f, j, c, g, k). During the phase transition, a large number of nano bubbles were generated (Fig. 3f, g and Supplementary Figs 2 and

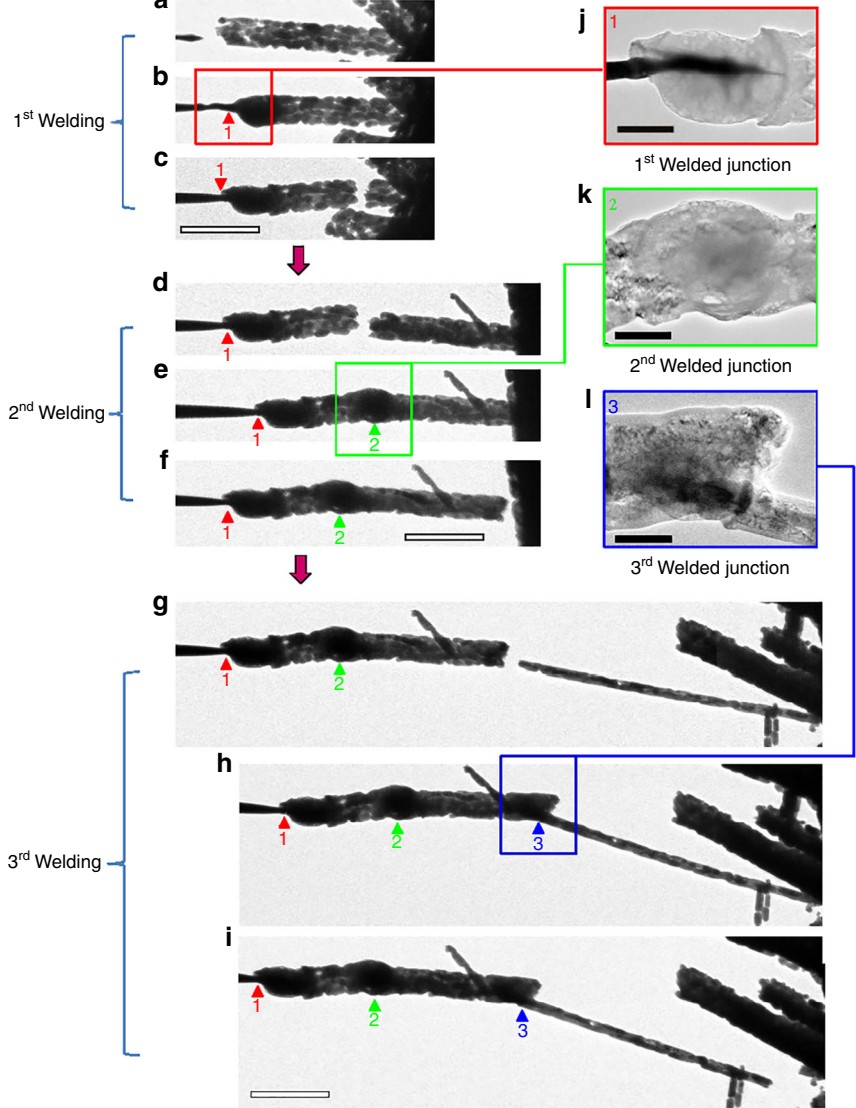

**Fig. 2** The welding process of the ceramic MgO nanowires. Arrowheads and the numbers underneath them point out the welding spots. **a–c** The first welding is the process of welding an MgO nanowire to the W tip. **a** A W tip approaching an MgO nanowire. **b** The MgO nanowire was welded to the W tip. **c** Tensile test on the MgO nanowire welded to the W tip. The MgO nanowire broke near the right contact. **d–f** The second welding: **d** The first nanowire welded to the W tip approaching a second MgO nanowire. **e** The second MgO nanowire welded together with the first one. **f** The second nanowire broke near the right contact upon tensile loading. **g–i** The third welding: **g**, the first and second welded nanowires approaching a third nanowire; **h** The nanowire welded to a third nanowire; **i** The third nanowire broke near the right contact upon tensile loading. **j, k, l** Magnifications of the boxed region in **b**, **e** and **h**, respectively, showing the morphologies of the welded spots. Scale bars: (**a–i**) 5 μm; (**j–l**) 200 nm

5). These tiny bubbles gradually merged into large bubbles and migrated quickly toward the nanowire edge (Fig. 3g), while new bubbles were continuously generated, forming a porous structure in the nanowires (Fig. 3c, g). After 16 minutes of reaction, the sharp angular edges of the two MgO nanowires (Fig. 3b) became blurred and the two nanowires were welded together (Fig. 3c, g and Supplementary Fig. 9). EDPs (Fig. 3i–k) and EELS (Fig. 3m, n) indicate that the reaction product in the welding spot was amorphous $MgCO_3$. With the presence of $CO_2$, the $MgCO_3$ obtained immediately after reaction was similar to a glue, and could be significantly stretched, displaying superplasticity (Supplementary Fig. 10 and Supplementary Movie 3). However, with increasing time, it solidified with many nanobubbles embedded in it (Supplementary Fig. 11). E-beam has an important effect on the mechanical behavior of the amorphous $MgCO_3$. With intense e-beam irradiation, the as-formed $MgCO_3$ was very ductile and showed a ductile fracture feature; when the beam was turned off, it became brittle (Supplementary Fig. 11).

Voids existing in the welding junction is detrimental to the mechanical strength of a weld (Supplementary Fig. 12 and Supplementary Movie 4). Thus, after finishing the welding (Supplementary Movie 5), the $CO_2$ was pumped out of the ETEM chamber to drive out the bubbles (Fig. 3d and Supplementary Movie 6). Under e-beam irradiation without the $CO_2$ gas, the nanobubbles in the amorphous $MgCO_3$ gradually disappeared (Fig. 3d, h), and the original porous, amorphous $MgCO_3$ (Fig. 3c, g, k) gradually transformed into a dense nanocrystalline material (Fig. 3h, Supplementary Fig. 13 and Supplementary Movie 6). The EDP (Fig. 3l) and the EELS results (Fig. 3m, n) indicated that these tiny nanocrystals were MgO (JCPDS card No. 30-0794), which originated from the decomposition of $MgCO_3$. Although the nanocrystalline MgO generated after welding had the same elemental composition as the pristine MgO, the grain size of the former was much smaller than the latter. It is well known that nanocrystalline materials usually demonstrate excellent mechanical properties[23, 24], thus ensuring the excellent mechanical properties on the weld junction, which has already been confirmed from Fig. 2. The above welding technology can be extended to MgO nanosheets (Supplementary Fig. 14) and CaO nanoparticles (Supplementary Fig. 15).

The most interesting phenomenon in this study is the melting behavior of MgO. It is well known that the melting point of ceramic MgO is very high (2852 °C), however, it showed a melting behavior under the e-beam irradiation in the $CO_2$ environment without any external heating. The melting behavior is not directly the melting of MgO but is associated with a chemical reaction between the MgO and the $CO_2$, i.e.

$$MgO + CO_2 \rightarrow MgCO_3 \tag{1}$$

Reaction (1) is exothermic with $\Delta H = -118\ kJ/mol$[25]. As MgO and $MgCO_3$ are superior thermal insulating materials, most of the released heat from this reaction can be completely absorbed, which can raise the temperature of $MgCO_3$ (1 mol) from room temperature to 1560 K ($\Delta T = \Delta H/C = 118000/75.6 = 1560\ K$) by referring to its specific heat capacity: $C = 75.6\ J/(mol\ K)$. The melting point of $MgCO_3$ is only 873–1173 K. Thus, it is understandable that $MgCO_3$ demonstrated an obvious melting behavior. Under normal conditions, there exists an energy barrier for the MgO to react with $CO_2$, and this reaction cannot occur automatically, but in this study, reaction (1) was activated by the e-beam irradiation. After evacuating the $CO_2$ and under e-beam irradiation, $MgCO_3$ decomposed back into MgO via the following

reaction:

$$MgCO_3 \rightarrow MgO + CO_2 \tag{2}$$

Thus in the whole welding process, $MgCO_3$ only played a transitional role. We reduced the melting temperature of the solder material through the carbonation of MgO, realizing a true ceramic nanowelding by using ceramic without any other external heat or current.

E-beam and $CO_2$ are the two prerequisites for realizing the ceramic nano welding. Both $CO_2$ and MgO were activated under high dose e-beam irradiation. For the $CO_2$, the predominant ions formed on irradiation are $CO_2^+$, $CO^+$, CO, $O^+$, $O_2^+$, $C^+$ etc[26]. While for the MgO, the Mg atoms and MgO molecules in the irradiated area are charged positively as $Mg^+$ and $(MgO)^+$, respectively[27]. Due to the formation of highly reactive products as a result of the radiolysis process, the reaction between MgO and $CO_2$ occurred in the ETEM. No external heat source or current are needed for the welding.

**Practical applications of the ceramic nanowelding technology.** Nanomaterials such as nanowires are building blocks of future electronic devices, and evaluating their mechanical properties is crucial but challenging. For example, measuring the tensile strength of individual nanowires is very challenging due to the difficulty in making robust mechanical contact to fix the nanowires to a testing device[28]. One way to fix the ceramic nanomaterials on a TEM chip for in situ test is through FIB manipulation[29]. Although the recent innovative cold welding technique displays a good performance for the welding of metal nanowires[14], it is unsuitable for the welding of nano ceramics.

In this study, we completed the welding of ceramic nanomaterials through a chemical reaction, in which the typical ceramic MgO was used as the solder. For evaluating the detailed mechanical properties of the weld junction, we created a self-assembled in situ TEM-atomic force microscopy (AFM) device to carry out the tensile test, as illustrated in Fig. 4a. The real image of the experimental setup is shown in Fig. 4b. In order to study the mechanical properties of the pure MgO nanocrystals on the weld junction, we first welded the W tip coated with some pure MgO nanocrystal onto the tip of the AFM cantilever. When stretching the W tip, the weld junction comprised of nanocrystalline MgO moved forward along the same direction, while the AFM cantilever was bent (Supplementary Fig. 16). After stretching 363 nm, the nanocrystalline MgO junction broke and the fracture surface displayed a brittle fracture characteristic (Supplementary Movie 7). The detailed tensile strain value of nanocrystalline MgO can be calculated through measuring the deflection of the AFM cantilever. According to the formula $F = K \cdot \Delta X$, where F is the force, K is the force constant, and $\Delta X$ is the displacement of the AFM cantilever, the tensile strength of the MgO nanocrystals on the weld junction reached as high as 2.8 GPa. This value is much higher than that of most of the welds using conventional solder materials, such as metals (Cu:~210 MPa, Iron:~350 MPa, Al:~40 MPa)[30] and silver glue (~20 MPa), which demonstrates its superior advantages for welding. Additionally, after the tensile test, we flowed $CO_2$ gas into the ETEM chamber and attempted to weld the fractured welding junction, and the fractured welds were fully rewelded (Supplementary Fig. 17), showing a great potential of the application of the current welding technology in the repairing of ceramic materials.

This ceramic nanowelding technology can enable in situ TEM tensile testing of other nanowire systems. For instance, we have firstly tested the tensile properties of ceramic MgO nanowires by using this technique (Fig. 4c). In the experiment, one pristine

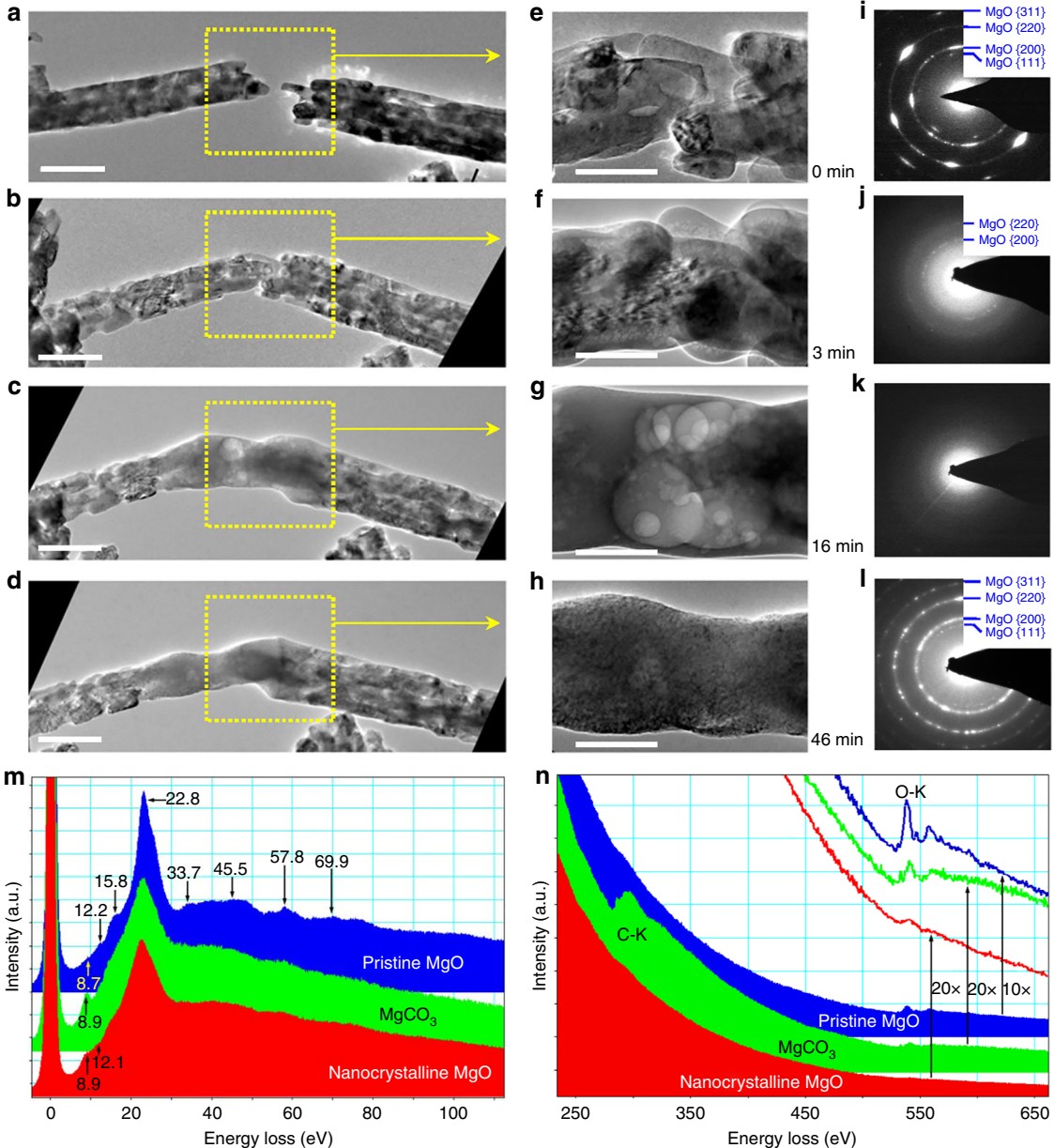

**Fig. 3** Structure evolution of the nanowelding process. **a**, **e** Two MgO nanowires brought into close proximity prior to the welding. **b**, **f** Upon connecting the two MgO nanowires, $CO_2$ is pumped into the ETEM chamber. Under e-beam irradiation in a $CO_2$ environment, the MgO carbonation reaction starts immediately. **c**, **g** The two MgO nanowires welded together. Plenty of voids were found in the weld. **d**, **h** After removing $CO_2$ from the ETEM chamber and continuing irradiation on the welded spot, the microstructure of the weld spot changes from grey amorphous contrast to nanocrystalline. **c**, **g** Plenty of nano-pores exist in the MgO after irradiation in the $CO_2$ environment for 3 min. The sharp diffraction rings from the pristine MgO (**i**) become dim, and the amorphous content gradually increases (**j**, **k**). **h** The bubbles in the $MgCO_3$ gradually disappear, and in the end, the original porous, amorphous $MgCO_3$ transforms into a dense nanocrystalline MgO (**l**). **m**, **n** Low-loss and core-loss EELS from the pristine MgO, $MgCO_3$ and the nanocrystalline MgO originate from the decomposition of $MgCO_3$. The pristine MgO shows seven characteristic low-loss peaks at 12.2, 15.8, 22.8, 33.7, 45.5, 57.8 and 69.9 eV. The peaks at 532 and 538 eV arise from the O-K edge. Note that C is only present in the $MgCO_3$ (green profile in **n**), which is not found in the pristine MgO (blue plot in **n**) and the nanocrystalline MgO (red plot in **n**). Scale bars: (**a–d**) 500 nm; (**e–h**) 200 nm

MgO nanowire was first welded onto the W tip, and then this nanowire was welded to another MgO nanowire with a much larger diameter. After they were firmly welded together, we pulled the W tip backward, and the entire sample moved along the tensile direction. Due to the ultrahigh mechanical strength of the weld junction, the MgO nanowire with a small diameter broke in the middle (Fig. 4c and Supplementary Movie 8). In this process, the W tip moved 1.05 µm. Referring to the area of the fracture surface, we calculated that the tensile strength of the MgO

nanowire to be 261 MPa (much lower than that of the nanocrystalline MgO).

In addition to MgO nanowires, we also carried out several tensile experiments for single crystal CuO (Fig. 4d, Supplementary Figs 18 and 19 and Supplementary Movie 8) and single crystal $V_2O_5$ nanowires (Supplementary Figs 20 and 21 and Supplementary Movie 8) in the ETEM by using this technique. It is calculated that the tensile strength of the CuO nanowire is 2.3 GPa, and that of the $V_2O_5$ nanowire is 1.6 GPa. It should be noted

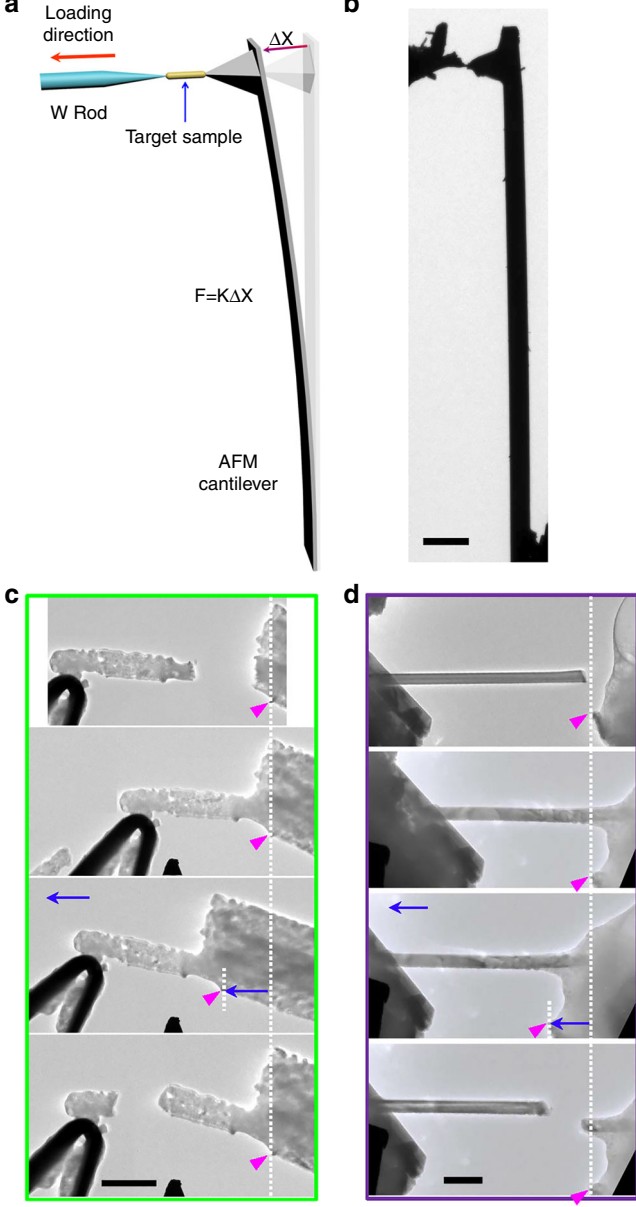

**Fig. 4** In situ tensile tests of ceramic nanowires using ceramic nanowelding. **a** The schematic of a self-assembled in situ TEM-AFM device for the tensile test. The target sample was first welded to the W tip, and then welded to the tip of the AFM cantilever. We stretched the W tip, and the welded sample moved along the same direction, meanwhile the AFM cantilever was bent. Through measuring the deflection of the AFM cantilever, we can calculate the tensile strength of the sample. **b** A real image of the experiment setup. **c** The tensile test for the pristine MgO nanowires, and the maximum moving distance of the W tip is 1 μm. The weld junction is composed of nanocrystalline MgO. **d** The tensile test for a ceramic CuO nanowire, and the maximum moving distance of the W tip is 228 nm. Scale bars: (**b**) 10 μm; (**c**) 1 μm; (**d**) 200 nm

that both the tested CuO and the $V_2O_5$ nanowires are single crystals (Supplementary Figs 18 and 20), and they both broke from locations other than the welding spots during the tensile test experiments, proving unambiguously that the strength of the welds is stronger than that of the nanowires. It also proves the validity of the current welding technique in joining single crystal nanowires for tensile testing experiments. Besides evaluating the mechanical properties of nanomaterials in the TEM, this

technique shows a great potential for the assembling of nanodevices, especially for fabricating or repairing ceramic nanodevices. In the future, it is possible that we could install such a device in the FIB equipment for achieving ceramic nanowelding using ceramic solder.

Similar to the welding of nanoceramic material, how to weld macroscopic ceramic is also a quite important but difficult task in industry. Inspired by the above experiment, we also explored the possibility of welding bulk ceramic materials using this technique via a macroscopic ceramic welding setup (Supplementary Fig. 22). In this study, we tried to weld ceramic $SiO_2$ fibers onto a Si wafer, which are widely used materials in semiconductor industry. It is known that plasma can be formed from gas molecules under e-beam irradiation, thus we generated a $CO_2$ plasma atmosphere in the chamber to promote the carbonation of MgO (Supplementary Fig. 23). The pristine MgO nanowires used for the macroscopic experiment were the same nanowires used in the in situ ETEM study. It was found that the MgO nanowires displayed a similar melting behavior as that observed in the ETEM (Supplementary Fig. 24). We found that a macroscopic ceramic $SiO_2$ fiber was firmly welded onto the Si wafer by using this technique (Supplementary Fig. 25 and Supplementary Movie 9). A fast chemical reaction between $CO_2$ and MgO at the macroscopic level took place in the chamber, similar phenomenon and byproduct was found as that observed in the ETEM study. The realization of plasma-assisted $CO_2$ reaction with MgO at a macroscopic scale has important practical significance for the ceramic industry, as welding ceramics using ceramic solder becomes possible.

## Discussion

A novel ceramic nanowelding technology based on the chemical reaction $MgO + CO_2 \rightarrow MgCO_3$ induced by e-beam irradiation is reported. The as-formed $MgCO_3$ decomposes to nanocrystalline MgO with release of $CO_2$ gas under e-beam irradiation. The formation of dense nanocrystalline MgO on the weld junction contributes to the strong mechanical properties of the rejoined nanowire. The mechanical strength of the weld junction can reach over 2.8 GPa. The technology can be used to weld ceramic nanowires and to enable in situ tensile test of individual nano-wires. The technology can be used in the welding of not only nanoscale ceramics but also macroscopic ceramic material, showing a great application potential of this technology to industry.

## Methods

**The welding of ceramic nanomaterials**. MgO nanowires were synthesized by a hydrothermal method, and the detailed fabrication process is discussed in the supporting information method 1. The TEM samples were prepared by adhering MgO nanowires onto W or Al STM probes and then loading the probes into the TEM-STM (Pico Femto FE-F20). The movement of the sample was manipulated by the piezo-electric tube of the holders. When the nanowires were connected, high-purity $CO_2$ (99.99%) was introduced into the specimen chamber with a pressure of 200 Pa. Upon the presence of $CO_2$ and e-beam irradiation, a welding process started immediately. When the welding interfaces vanished, we pumped out the $CO_2$ from the ETEM chamber and continued with e-beam irradiation of the nano welds. After that we found that the microstructure of the welds turned from amorphous $MgCO_3$ into nanocrystalline MgO. In order to quantitatively measure the mechanical strength of the weld junctions and carry out tensile tests for some other ceramic nanowires, we inserted a silicon AFM cantilever beam ($K = 40$ N m$^{-1}$) into one end of the TEM-STM holder which could be used as a TEM-AFM device. Because the deflection of the cantilever was much smaller than its beam length, a linear relationship between DD (displacement of the AFM tip, equal to the cantilever deflection) and F (force applied on the nanowire sample) was assumed. Attachment of the sample to the AFM cantilever was carried out using this welding technique and MgO was selected as the solder. During the experiment, a beam blocking bar was inserted into the field of view as the reference for displacement measurements. The tensile strength was calculated as engineering stress. The stress calculation was reasonably accurate ($< + 10\%$ error) by measuring cantilever deflections in high magnification TEM images. The in situ welding experiment was

conducted in a Cs-corrected environmental TEM (FEI, Titan ETEM, 300 kV). During the welding experiments, no current was passed through the sample, and the intensity of the e-beam was 0.02~0.2 A cm$^{-2}$.

**The welding of bulk ceramic materials**. After the success of welding ceramic nanomaterial, we attempted this ceramic welding technique on macroscopic ceramics. Hydrothermal synthesized MgO powders were the same as that used for the TEM, and were put into a home-made glow discharge chamber. Upon the welding process, $CO_2$ gas in high-purity (99.99%) was introduced into the chamber and kept at ~200 Pa for the generation of plasma. The discharge voltage was 500 V, current 110 mA, electrode space ~1 cm and treatment time was ~1 h.

**Data availability**. The authors declare that all data supporting the findings of this study are available within the Article and its Supplementary Information files. Any other data will be provided by the corresponding author upon request.

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

## Acknowledgements

This work was financially supported by the National Natural Science Foundation of China (Nos. 51772262, 21576289, 51401239, 51071175, 51231005, 11575154 and 51621063), National Key Research and Development Program of China (No. 2017YFB0702001), the Science Foundation of China University of Petroleum, Beijing (No. C201603), the High-Level Talents Research Program of the Yanshan University (grant number 005000201) and Thousand Talents Program.

## Author contributions

J.H., Z.S. and L.Z. conceived and designed the project. Y.T. and F.Y. fabricated the samples. L.Z., Y.T., Y.W., Q.L., T.Y., C.D. and Y.S. carried out the in situ ETEM experiments. J.H., T.S., L.C. and Q.P. supervised the experiments. J.H., L.Z., Y.L., and Z.S. co-wrote the paper. L.Z. and Q.P. contribute equally to this work. All the authors discussed the results and commented on the manuscript.
