## [Peer Review File · Nature Communications]

Reviewer #1 (Remarks to the Author):

The authors report a new technique to weld nanowires with a ceramic joint, as a result of a chemical reaction occurring in an environmental transmission electron microscope. They synthesized a ceramic nanowelding material through carbonation of the MgO nanowire under the electron beam, without using external heat or current, producing nanocrystalline MgO as resulting material joint which, upon tensile testing, appears to be stronger than the MgO nanowire itself. The work is interesting and provides new insights for the welding technology of oxide ceramics. I recommend publication, with a minor revision:

I have one comment regarding an acronym: In p.3, paragraph 2, line 4 the authors write scanning transmission microscopy (STM) to refer to the sample holder. I guess they mean scanning tunneling microscopy.

Reviewer #2 (Remarks to the Author):

This manuscript by Zhang et al. presents a novel in situ method to weld several oxide nanowires by using porous MgO as solder and under CO₂ environment, generating welding spots with strengths comparable to the welded nanowires. The authors suggest that the present method has a wide applicability not only to the bottom-up building of ceramic nanodevices, but also to the macroscopic welding in ceramic industry.

The results are valuable, the testing and characterization carried out and presented well; however, I think the authors should address the following issue clearly to make their claims convincing, i.e., what is the role (pros and cons) of the porous structure of the nanowires focused here? It seems that the high specific area of the porous nanowires makes them easily wetted (and permeated) by the MgCO₃ fluid, but the high porosity is also responsible for the relatively low strength of the "pristine" nanowires, especially compared to typical single-crystalline ceramic nanowires. Therefore, the authors are prompted to provide a clear evidence that the present method can also weld a single-crystalline nanowires (SiC, ZnO, etc., perhaps the CuO nanowire shown here?) with a satisfactory strength of welding spots, i.e., comparable to the welded nanowires, before this manuscript can be considered for publication.

Reviewer #3 (Remarks to the Author):

Referee report for manuscript- NCOMMS-17-18596

High-quality joining of ceramics is a bottleneck challenge that limits the applications of high-performance ceramics in the vast of industrial applications. To address this issue, the authors have proposed a rather novel and efficient approach, named as cold welding, in which the energetic beam irradiation assisted chemical reaction in the presence of CO₂ gases plays the key role, and the whole process was performed inside a state-of-art environmental TEM. Successful examples on connecting MgO, CuO and V₂O₅ nanowires were tested in situ, and the re-joined nanowires exhibit remarkable mechanical properties as evidenced by tensile tests. The authors further explored the potential applications of this method on macroscopic welding on other ceramic materials such as SiO₂. All these results have clearly collaborate the promising future of this novel method on the welding of ceramics in the future. As such, the present referee would like to recommend its publication in Nature Communications.

While the present form is still far from mature, I would suggest the authors consider the following issues in their revised manuscript.

1. Can the authors give more explanations on the reasons why the re-joined nanowires are stronger in terms of mechanical properties than the pristine one?

Is that mainly due to the polycrystalline or highly defective nature of the chosen nanowires for the

welding experiments? Does the contact strong enough and remains unchanged during the whole test? An ideal candidate material would be the single crystalline nanowires. Major point.

2. Is there limits in terms of diameter for the cold welding? Minor point

3. Most important issue-beam effect.

i) It is well know that metal oxides may undergo a radiolysis process upon the illumination of high-energy electron beam, and sometimes leads to the formation of metal, such as metal-copper. The authors mostly focus on the direct reaction of metal oxides (MgO, CuO) with CO₂, while pay less attention to the former one. During the whole process, if metal, say Cu was formed due to the beam-damage, and then the welding could be easily realized through Cu-Cu metalling bonding, rather than via the beam assisted chemical reaction between CuO and CO₂. And then, such a metallic junction may be further oxidized in the presence of surround oxides (may serve as the source oxygen during the beam damage process) and the CO₂ gases. This possibility of this procedurd should be carefully examined and discussed. By the way, the cold welding of metallic nanowires via in-situ TEM was also demonstrated previous by one of the present authors.

ii) As the electron beam plays an indispensable role during the whole welding, it is necessary to quantify the contribution of electron dose. Is there any dose dependence there?

iii) Interaction of energy electron beam with CO₂ gases will induce the ionization process, and lead to the formation of highly reactive products as a result of radiolysis process, particularly close to the sample surface. It should also be addressed in the discussion.

Reviewer #1:

The authors report a new technique to weld nanowires with a ceramic joint, as a result of a chemical reaction occurring in an environmental transmission electron microscope. They synthesized a ceramic nanowelding material through carbonation of the MgO nanowire under the electron beam (e-beam), without using external heat or current, producing nanocrystalline MgO as resulting material joint which, upon tensile testing, appears to be stronger than the MgO nanowire itself. The work is interesting and provides new insights for the welding technology of oxide ceramics. I recommend publication, with a minor revision: I have one comment regarding an acronym: In p.3, paragraph 2, line 4 the authors write scanning transmission microscopy (STM) to refer to the sample holder. I guess they mean scanning tunneling microscopy.

Response: We thank the referee for his/her positive comments on our manuscript! Regarding the acronym STM, it indeed means scanning tunneling microscopy, and we have corrected this error in the revised manuscript. We appreciate the referee for pointing out this error.

Reviewer #2:

This manuscript by Zhang et al. presents a novel in situ method to weld several oxide nanowires by using porous MgO as solder and under CO₂ environment, generating welding spots with strengths comparable to the welded nanowires. The authors suggest that the present method has a wide applicability not only to the bottom-up building of ceramic nanodevices, but also to the macroscopic welding in ceramic industry.

The results are valuable, the testing and characterization carried out and presented well; however, I think the authors should address the following issue clearly to make their claims

convincing, i.e., what is the role (pros and cons) of the porous structure of the nanowires focused here? It seems that the high specific area of the porous nanowires makes them easily wetted (and permeated) by the $MgCO_3$ fluid, but the high porosity is also responsible for the relatively low strength of the "pristine" nanowires, especially compared to typical single-crystalline ceramic nanowires. Therefore, the authors are prompted to provide a clear evidence that the present method can also weld single-crystalline nanowires (SiC, ZnO, etc., perhaps the CuO nanowire shown here?) with a satisfactory strength of welding spots, i.e., comparable to the welded nanowires, before this manuscript can be considered for publication.

Response: We thank the referee for his/her constructive suggestions. Herein, we report a novel technique to weld ceramic nanowires with mechanical strength of the welding spots even stronger than that of the pristine nanowires. The referee suggests that we should provide clear evidence that our method is also applicable to single crystalline nanowires. This is an excellent point, as conventional wisdom tells us that a porous structure usually possesses lower mechanical strength than a solid structure. As the referee suggested, we indeed performed welding experiments on single crystalline nanowires with no pores, and the conclusion remains the same, namely the welding spots are so strong that the single crystal nanowires all broke from the nanowires rather than from the welding joints. We conducted welding and then tensile test experiments on many single crystalline CuO and V_2O_5 nanowires, and representative results are presented in Fig. 4d, and Supplementary Figs. 19, 21. TEM and HRTEM images indicate that both the CuO (Fig. R1) and V_2O_5 (Fig. R2) nanowires are single crystals. Tensile tests experiments indicated that the fracture strength of the single crystal CuO and V_2O_5 nanowires is 2.3 GPa and 1.6 GPa, respectively. We have also conducted extensive tensile tests on the mechanical strength of the welding spots, and the results showed that the fracture strength of the welds can reach as high as 2.8 GPa (Supplementary Fig. 16, Video 7). In the meantime, the fact that all nanowires broke from locations other than the welding spots (Fig. 4d, Supplementary Figs. 19, 21) proves that the mechanical strength of the welding spots is stronger than that of the single crystalline CuO and V_2O_5 nanowires. It should be noted that the above results were presented in our original manuscript, however, we did not point out that the tested CuO and V_2O_5 nanowires are single crystals. We apologize for this missing information, which is now provided in our revised manuscript (line 240, page 8).

Fig. R1 (a) A TEM image of a CuO nanowire prepared by heating a copper grid at 500 °C for 2 h. (b, d) High-resolution TEM images showing the single crystal structure of a nanowire from two different zone axes. (c, e) Fast Fourier Transformation (FFT) patterns of the framed areas in (b, d), respectively. These results indicate that the CuO nanowire is a single crystal.

Fig. R2 (a) A TEM image of a V_2O_5 nanowire prepared by a hydrothermal method. (b) A high-resolution TEM image obtained from the edge of the V_2O_5 nanowire. (c) A high-resolution TEM image showing the single crystal structure of the nanowire. (d) The FFT pattern of the framed area in (b). These results confirmed that V_2O_5 nanowire is a single crystal.

In summary, the welding technology presented in our manuscript is not restricted to porous MgO nanowires but applicable to single crystalline nanowires such as CuO and V_2O_5 nanowires as well, indicating the broad applications of this novel welding technology.

Reviewer #3 (Remarks to the Author):

Referee report for manuscript- NCOMMS-17-18596

High-quality joining of ceramics is a bottleneck challenge that limits the applications of high-performance ceramics in the vast of industrial applications. To address this issue, the authors have proposed a rather novel and efficient approach, named as cold welding, in which the energetic beam irradiation assisted chemical reaction in the presence of CO₂ gases plays the key role, and the whole process was performed inside a state-of-art environmental TEM. Successful examples on connecting MgO, CuO and V₂O₅ nanowires were tested in situ, and the re-joined nanowires exhibit remarkable mechanical properties as evidenced by tensile tests. The authors further explored the potential applications of this method on macroscopic welding on other ceramic materials such as SiO₂. All these results have clearly collaborated the promising future of this novel method on the welding of ceramics in the future. As such, the present referee would like to recommend its publication in Nature Communications.

While the present form is still far from mature, I would suggest the authors consider the following issues in their revised manuscript.

1. Can the authors give more explanations on the reasons why the re-joined nanowires are stronger in terms of mechanical properties than the pristine one? Is that mainly due to the polycrystalline or highly defective nature of the chosen nanowires for the welding experiments? Does the contact strong enough and remains unchanged during the whole test? An ideal candidate material would be the single crystalline nanowires. Major point.

Response: We thank the referee for his/her recognition of our work and very positive comments on our manuscript. Referee 3's main suggestion is essentially similar to that of referee 2: that is the strong welding may originate from the porous MgO nanowire, therefore we should conduct welding and tensile test experiments in single crystalline nanowires to demonstrate that the mechanical strength of the welding spot is stronger than that of the pristine single crystalline nanowires. This is an excellent suggestion. As we already addressed in our response to referee 2, we indeed conducted welding and tensile test experiments on single crystal CuO and V₂O₅ nanowires. The results reinforce our conclusion that the mechanical strength of the welding spots is stronger than that of the pristine single crystalline nanowires, as both the single crystalline CuO and V₂O₅ nanowires broke from the nanowires rather than from the welding joints (Fig. 4d,

Supplementary Figs. 19, 21).

Regarding to referee 2's question: "*...the reasons why the re-joined nanowires are stronger in terms of mechanical properties than the pristine one?*", the answer is the following: in our ceramic welding experiments, the process involves two steps: (1) $\text{MgO} + \text{CO}_2 \rightarrow \text{MgCO}_3$. After this chemical reaction, the original MgO is transformed into fluidic MgCO_3 , which can connect nanowires like a glue; (2) $\text{MgCO}_3 \rightarrow \text{MgO} + \text{CO}_2$. After the formation of MgCO_3 , CO_2 was pumped out of the ETEM chamber. Without CO_2 and under e-beam irradiation condition, the as formed MgCO_3 gradually transformed into nanocrystalline MgO plus CO_2 , and the CO_2 was released to the high vacuum of the ETEM. Consequently, a dense nanocrystalline MgO with an excellent mechanical property formed. It is because of the formation of the dense nanocrystalline MgO on the weld joints that contributes to the strong mechanical properties of the rejoined nanowire.

It is well known that nanocrystalline materials generally demonstrate superior mechanical properties as compared to its bulk counterpart due to the well-known Hall-Petch strengthening mechanism. The same principle may apply to the ceramic MgO welding joints, which showed remarkable mechanical strength, which explains why the mechanical strength of the welding spots is even stronger than the pristine nanowires. We recently conducted extensive tensile tests on the nanocrystalline MgO formed in the weld junction, and found that the fracture strength of the MgO nanocrystal junctions can reach as high as 2.8 GPa (Fig. R3), indicating the welding joints is indeed very strong.

In response to the question: "*Does the contact strong enough and remains unchanged during the whole test?*", the answer is yes. We did not see changes or slippage of the contact during all the tensile testing experiments.

Fig. R3 The in situ tensile tests for the MgO nanocrystals on the weld junction. (a, b) A welding junction was formed between a W tip (on the left) and an AFM cantilever (on the right). (c, d) The welding junction was then pulled towards the left direction until it fractured. The tensile strength for this particular weld was 2.8 GPa.

2. Is there limits in terms of diameter for the cold welding? Minor point

Response: There appears to be no limits in terms of the diameter for welding of nanowires. In our experiments, MgO is only used as the solder and there is no restriction on the size of the target nanowires. However, the bigger the nanowires, the longer it takes to weld the nanowires. As shown in the Fig. R4, both MgO nanowires with small (Figs. R4 a,b) and large diameters (Figs. R4 c,d) can be welded by using this technique. Moreover, we can even weld macroscopic ceramic materials by using this method (Figs. R4 e,f).

Fig. R4 Samples with different diameters are welded. (a) and (b), (c) and (d), (e) and (f) are MgO nanowires with different diameters before and after welding, respectively.

3. Most important issue-beam effect.

i) It is well known that metal oxides may undergo a radiolysis process upon the illumination of high-energy e-beam, and sometimes leads to the formation of metal, such as metal-copper. The authors mostly focus on the direct reaction of metal oxides (MgO, CuO) with CO₂, while pay less attention to the former one. During the whole process, if metal, say Cu was formed due to the beam-damage, and then the welding could be easily realized through Cu-Cu metalling bonding, rather than via the beam assisted chemical reaction between CuO and CO₂. And then, such a metallic junction may be further oxidized in the presence of surround oxides (may serve as the source oxygen during the beam damage process) and the CO₂ gases. This possibility of this procedure should be carefully examined and discussed. By the way, the cold welding of metallic nanowires via in-situ TEM was also demonstrated previously by one of the present authors.

Response: We agree that e-beam irradiation does cause a certain degree of irradiation damage to materials, but the welding of two CuO nanowires only through e-beam irradiation without CO₂ gas was not possible in our experiments. As shown in Fig. R5, two CuO nanowires were connected in the ETEM. After focusing the e-beam to the connection junction for 30 minutes with a high dose rate (~900 e/nm²·s) without CO₂ gas, welding did not take place between the two nanowires. Therefore, the proposed cold welding mechanism did not apply to our experiments (Fig. R5).

It has been reported that Ag-Ag, Au-Au and other metal materials can be welded together by cold welding, but they are all metallic materials and their diameters are usually rather small [Nat. Mater. 11, 241-249 (2012); Nat. Nanotechnol. 5, 218 (2010)]. While in this work, we can realize the welding of ceramic materials including MgO, V₂O₅, CuO by using MgO as the solder through a typical chemical reaction, which is completely different from the cold welding techniques as proposed by the referee.

Although the target ceramic nanomaterials with different types and sizes can be welded together by using our technique, it is found that only MgO and CaO can be used as the solder for this ceramic welding (Fig. R6 a,b). As shown in Fig. R6 c,d, CuO and ZnO nanowires demonstrate almost no changes under the e-beam irradiation combined with the existence of CO₂.

Fig. R5 In situ welding experiments of CuO nanowires by e-beam irradiation without CO₂ gas. (a) The pristine two CuO nanowires. (b) The two CuO nanowires are connected together by the movement of the W tip. (c) After focusing the e-beam to the connect junction (red circle) for 30 min ($\sim 900 \text{ e/nm}^2 \cdot \text{s}$), almost no change on the junction part was observed. (d) The two CuO nanowires are separated, proving that no welding occurred.

Fig. R6 The melting behavior of different types of ceramic nanomaterials under the e-beam irradiation in a CO₂ environment. Significant changes are found for (a) MgO and (b) CaO nanowires under the e-beam irradiation with the existence of CO₂. Almost no changes are observed for (c) CuO and (d) ZnO nanowires under the same condition.

ii) As the e-beam plays an indispensable role during the whole welding, it is necessary to quantify the contribution of electron dose. Is there any dose dependence there?

Response: E-beam dose rate plays a significant role in the welding process (Fig. R7). When the e-beam is blank, no reaction occurred between the MgO and CO₂ (Fig. R7a). By increasing the e-beam dose rate to about 100 e/nm²·s, the reaction proceeds with medium-speed (Fig. R7b). When the e-beam irradiation dose rate reaches about 870 e/nm²·s, the high-speed reaction quickly leads to formation of large amount of highly mobile bubbles in the interior of the irradiated MgO nanowire (Fig. R7c). This is addressed in our revised manuscript (line 88, page 3).

Fig. R7 E-beam dose rate plays a significant role in the welding process. (a) When the e-beam is blank, no reaction occurred between the MgO and CO₂. (b) By increasing the e-beam dose rate to about 100 e/nm²·s, MgO reacted with CO₂ with medium-speed. (c) When the e-beam irradiation dose rate reaches as high as 877 e/nm²·s, a large amount of highly mobile bubbles emerged in the interior of the irradiated MgO nanowire.

iii) *Interaction of energy e-beam with CO₂ gases will induce the ionization process, and lead to the formation of highly reactive products as a result of radiolysis process, particularly close to the sample surface. It should also be addressed in the discussion.*

Response: Interactions between high energy electrons with gas leads to ionization of gas molecules. The ionized molecules are very active and react with samples. In fact, both CO₂ and MgO can be activated under the e-beam irradiation with a high electron dose rate. In the case of CO₂, the predominant ions formed on irradiation are CO₂⁺, CO⁺, CO, O⁺, O₂⁺, and C⁺ etc. [Can. J. Chem. 48, 1951-1954 (1970)]. While for the MgO nanowire, the Mg atoms and MgO molecules in the irradiated area can be positively charged as Mg⁺ and (MgO)⁺, respectively [Controlled Atmosphere Transmission Electron Microscopy Principles and Practice (Springer International Publishing, 2016)]. Due to the formation of highly reactive products as a result of radiolysis process, the reaction between MgO and CO₂ takes place quickly in the ETEM. No external heat source and current are

needed for the welding. We have added related discussions in the revised manuscript (line 188, page 6).

Reviewer #2 (Remarks to the Author):

It turns out that the concerns raised in previous review, especially the applicability of the welding technique to single-crystalline nanowires, are well addressed in the revised form. I thus recommend its publication in Nature Communications.

Reviewer #3 (Remarks to the Author):

The present referee has no further comments, and would like to recommend its publication in Nature Communications.